# Formation of Three-Dimensional Spheres Enhances the Neurogenic Potential of Stem Cells from Apical Papilla

**DOI:** 10.3390/bioengineering9110604

**Published:** 2022-10-22

**Authors:** Mohammed S. Basabrain, Jialin Zhong, Haiyun Luo, Junqing Liu, Baicheng Yi, Ahmed Zaeneldin, Junhao Koh, Ting Zou, Chengfei Zhang

**Affiliations:** 1Restorative Dental Sciences, Endodontics, Faculty of Dentistry, The University of Hong Kong, Hong Kong SAR, China; 2Stomatological Hospital, Southern Medical University, 366 Jiangnan Avenue South, Guangzhou 510280, China; 3Restorative Dental Sciences, Cariology, Faculty of Dentistry, The University of Hong Kong, Hong Kong SAR, China

**Keywords:** neurogenic potential, neurites, SCAPs, sphere, stem cells

## Abstract

Cell-based neural regeneration is challenging due to the difficulty in obtaining sufficient neural stem cells with clinical applicability. Stem cells from apical papilla (SCAPs) originating from embryonic neural crests with high neurogenic potential could be a promising cell source for neural regeneration. This study aimed to investigate whether the formation of 3D spheres can promote SCAPs’ neurogenic potential. Material and methods: Three-dimensional SCAP spheres were first generated in a 256-well agarose microtissue mold. The spheres and single cells were individually cultured on collagen I-coated μ-slides. Cell morphological changes, neural marker expression, and neurite outgrowth were evaluated by confocal microscope, ELISA, and RT-qPCR. Results: Pronounced morphological changes were noticed in a time-dependent manner. The migrating cells’ morphology changed from fibroblast-like cells to neuron-like cells. Compared to the 2D culture, neurite length, number, and the expression of multiple progenitors, immature and mature neural markers were significantly higher in the 3D spheres. BDNF and NGF-β may play a significant role in the neural differentiation of SCAP spheres. Conclusion: The formation of 3D spheres enhanced the neurogenic potential of SCAPs, suggesting the advantage of using the 3D spheres of SCAPs for treating neural diseases.

## 1. Introduction

Neural injuries and diseases can lead to severe physical and mental impairment, such as hypoesthesia, paralysis, and dementia, negatively affecting the patient’s life [1]. Injuries to neural tissue cause cell damage and necrosis, inflammation, ischemia, and cell signaling alteration, leading to demyelination, axonal degeneration, and an inhibitory environment for regeneration [2]. Various strategies have been investigated, including drug delivery, cell-based approaches, and tissue engineering; however, effective and predictable therapies are still lacking [3]. Stem cell therapy, including mesenchymal stem cells (MSCs), neural stem cells, neural progenitor cells (NPCs), embryonic stem cells, and induced pluripotent stem cells, is a promising treatment via supporting autologous neural proliferation or self-differentiation to replace the injured neural cells and immunomodulation [4]. However, preclinical and clinical studies on stem cell therapy only demonstrated limited benefits, suggesting that a combined treatment is needed [4,5].

Dental stem cells, which express multiple embryonic stem cell markers, including Oct-4, NANOG, SSEA-3, SSEA-4, etc., can differentiate into various tissue-specific cells [6]. In particular, dental stem cells may serve as promising cell sources for neural tissue regeneration since they share a common origin with neural crest cells and possess high neurogenic potential [7]. Compared to dental pulp stem cells (DPSCs), SCAPs proliferate 2- to 3-fold greater and have higher proliferative and regenerative capacities and telomerase activity [8,9]. SCAPs express neural markers, including Nestin, β-tubulin III, neuronal nuclear protein (NeuN), neurofilament protein (NFM), and neuron-specific enolase [8]. In addition, SCAPs secrete a sufficient amount of brain-derived neurotrophic factor (BDNF) and can stimulate neurite elongation in vitro and in vivo [10]. In this context, SCAPs are an attractive source for neural regeneration.

To achieve successful cell-based therapies, a substantial number of cells are required for transplantation, which necessitates extensive ex vivo cell expansion. However, long-term in vitro culture and serial passages using the conventional two-dimensional (2D) dish are quite likely to evoke continuous changes in MSCs. Stem cells may gradually lose their self-renewal ability, differentiate into bone cells or adipocytes spontaneously, and undergo senescence [11]. To overcome these deleterious effects and their possible consequent functional alterations, 3D cell culture has been employed in various studies, which can better simulate the growth state of cells in vivo, promote cell self-renewal, inhibit their differentiation, and maintain the original characteristics of cells [11,12]. For example, short-term spheroid formation of adipose-derived stem cells (ASCs) can enhance their stemness, angiogenic capacity, and chemotaxis and thereby increase their therapeutic potential for tissue regeneration [13]. Similarly, the expression of stemness markers, NANOG, TP63, and CD44 in the spheroids of human DPSCs was much higher than within the monolayer cultures. The expression of neural markers, NFM, and β-tubulin III in the spheroids was increased in a time-dependent manner. Intriguingly, cells in the spheroids spontaneously differentiated into neuron-like cells after 1-week cultivation without any neural induction in a serum-free medium [14]. Kim BC et al. also found that three-dimensional (3D) SCAP clusters strongly expressed neuronal maturation markers, such as NFM, compared to non-aggregated cells under neural inductive culture conditions [15]. Although many studies have been performed to investigate the neural differentiation of dental stem cells, the induction methods are highly variable. The most noticeable variables are the duration of neurosphere formation, the duration of induction, the medium used, etc. Nevertheless, the optimal conditions for high-yielding neural cells from dental stem cells are unmet. Therefore, this study aimed to investigate the effects of 2D vs. 3D cultures without neural induction on early cell fate, neurite outgrowth, and the differentiation potential of SCAPs.

## 2. Materials and Methods

Unless otherwise stated, chemical reagents were purchased from Sigma-Aldrich (St. Louis, MO, USA). Culture medium and supplements were purchased from Gibco-Invitrogen (Carlsbad, CA, USA). All plastic labware consumables were purchased from Becton-Dickinson (Franklin Lakes, NJ, USA).

### 2.1. SCAPs Isolation and Characterization

The study protocol for procuring SCAPs from human dental tissue was approved by the Institutional Review Board of the University of Hong Kong/Hospital Authority Hong Kong West Cluster (HKU/HA HKW IRB. IRB Reference Number: UW 21-378). After obtaining informed consent, SCAPs were isolated from the freshly extracted third molars of a 20-year-old female patient at the Oral and Maxillofacial Clinic in Prince Philip Dental Hospital. Briefly, the apical papilla was gently removed from the apex, and the SCAPs were isolated as previously described [9]. SCAPs were cultivated in α-minimum essential medium (α-MEM) containing 10% (*v*/*v*) fetal bovine serum (FBS) and 1% (*v*/*v*) penicillin-streptomycin antibiotic solution (ThermoFisher Scientific, Inc. Waltham, MA, USA) at 37 °C in a 5% CO2 incubator. The medium was changed every two days. SCAPs were characterized by flow cytometric analysis based on their expression of CD45, CD73, CD90, and CD105, as well as multi-lineage differentiation assays using osteogenic and neurogenic induction protocols [16,17,18] and chondrogenic and adipogenic protocols (GUXMX-90041, GUXMX-90031, Cyagen, Santa Clara, CA, USA) according to the manufacturer’s instructions. Passages 3 to 6 were used for these experiments.

### 2.2. Formation and Culture of 3D SCAP Spheres

Upon reaching 70 to 80% confluence, SCAPs were digested with 0.5% (*w*/*v*) trypsin-EDTA. Next, 2 × 10^5^ cells in 190 µL medium were transferred into each well of the 256-well agarose microtissue mold (MicroTissues, Inc, Sharon, MA, USA) and cultured for 3 days to generate spheres [19]. Then, 15–20 spheres were collected from the mold and seeded on the top of μ-slides (Cat. No. 80427, ibidi GmbH, Lochhamer Schlag, Gräfelfing, Germany) coated with rat tail collagen I (Cat. No. 354236, Corning, NY, USA). Alternatively, SCAPs were seeded in the 6-well plates for 3 days and then digested using 0.5% (*w*/*v*) trypsin-EDTA. The 2 × 10^5^ cells were cultured on the top of μ-slides coated with rat tail collagen I. The culture medium was α-MEM containing 10% (*v*/*v*) FBS and 1% (*v*/*v*) penicillin-streptomycin antibiotics. The medium was changed every 3 days. The samples were evaluated on days 4 and 7.

### 2.3. Live and Dead Assay and Immunofluorescence Staining

After seeding 3D and 2D SCAPs on collagen, cell viability was evaluated on the fourth and seventh days by LIVE/DEAD kit (Cat. No. L3224, Gibco-Invitrogen, Carlsbad, CA, USA). The assay was conducted following the manufacturer’s recommendation. The samples were rinsed with phosphate-buffered saline (PBS) and then treated with PBS containing 2 mmol/L ethidium homodimer-1 and 4 mmol/L calcein AM. After 30 min of the treatment, the samples were washed with PBS and assessed using confocal microscopy. The live cells had green fluorescence signals, while the dead cells carried a red signal.

The expression of neural markers (Nestin, β-tublin III, microtubule-associated protein-2 (MAP-2), and NeuN) were analyzed by immunofluorescence on days 4 and 7. Briefly, the cells were fixed with 4% (*v*/*v*) paraformaldehyde for 20 min, permeabilized with 0.1% (*w*/*v*) Triton X-100 for 10 min, and blocked in PBS with 1% (*w*/*v*) bovine serum albumin (BSA) for two hours. Then, the samples were incubated with primary antibodies at 4 °C overnight. The following antibodies were utilized: rabbit anti-MAP-2 (1:200 dilution) (Cat. No. AB5622-I Merck KGaA, Darmstadt, Germany), rabbit anti-NeuN (1:100 dilution) (Cat. No. 702022, Invitrogen, Carlsbad, CA, USA), mouse anti-β-tubulin III (1:500 dilutions) (Cat. No. AB78078, Abcam, Cambridge, UK), and mouse anti-Nestin (1:200 dilution) (Cat. No. 33475S, Cell Signaling Technology, Danvers, MA, USA). The excessive primary antibodies were washed with 10% PBS. The samples were then incubated with the corresponding secondary antibodies (1:200), i.e., goat anti-mouse secondary antibody conjugated to Alexa Fluor 488 (Cat No. ab150117, Abcam) or goat anti-rabbit secondary antibody conjugated to Alexa Fluor 594 (Cat No. ab6718, Abcam) for two hours in the dark at room temperature. After washing the excess secondary antibodies with 10% PBS, the samples were imaged under a laser scanning confocal microscope (LSM900, ZEISS, Jena, Germany) using a 10× objective lens under the specific excitation/emission wavelengths for Alexa Fluor 488 (490/580 nm) and Alexa Fluor 594 (565/700 nm).

### 2.4. Enzyme-Linked Immunosorbent Assay (ELISA)

Both 3D spheres and 2D cultures of SCAPs were cultivated in 6-well plates for another four days, and 5 × 10^5^ cells were used in both groups. The supernatants were collected and centrifuged at 1500 rpm (Centrifuge 5420; Eppendorf, Hamburg, Germany). BDNF and nerve growth factor-beta (NGF-β) were measured by the ELISA kits according to the manufacturer’s instructions (ab212166 and ab193760, Abcam).

### 2.5. Image Analysis for Live and Dead Assay, Markers, and Neurites

The quantification of the red and green signals in the live and dead assay was carried out using BioImage software (Free foundation software, Boston, MA, USA). Neurites were defined as a process extending from the cell body ≥ 20 µm [20]. The expression of Nestin, β-tubulin III, MAP-2, and NeuN was measured by the fluorescent density with ImageJ software using the following formula: integrated density = (area of selected cell × mean fluorescence of background readings). All the measurements were taken using NeuronJ plugins in ImageJ software (NIH Image, Bethesda, MD, USA). Six cells were randomly selected, which were out of the sphere and entirely within the frame of the image. The scale was set according to the reference scale within the image. The data collected from four images were used for statistical analysis. All the images were processed using ZEN Blue software (ZEISS, Germany).

### 2.6. RT-qPCR

The RNA from 2D and 3D SCAPs were extracted using TRIzol Reagent (Thermo Fisher Scienctific, Waltham, MA, USA) following the manufacturer’s instruction protocol with slight modification. To obtain sufficient spheres for gene isolation, we used 15 agarose molds, and to isolate the RNA from the cells in collagen, we increased the lysis time from 5 to 10 min. Afterward, reverse transcription was conducted with SuperScript VILO Master Mix (Life Technologies, Grand Island, NY, USA). RT-qPCR analysis was conducted with the StepOne Real-Time PCR System (Applied Biosystems, Grand Island, NY, USA) using SYBR Premix Ex Taq II (Cat No. RR820A, Takara Bio, Shiga, Japan). The neural primer sequences of octamer-binding transcription factor (Oct-4), SRY-box transcription factor 2 (Sox-2), Sox-10, Nestin, β-tubulin III, MAP-2, NeuN, neuron-specific enolase (NSE), NFM, BDNF, and NGF for the RT-qPCR analysis are shown in Table 1, with Glyceraldehyde 3-phosphate dehydrogenase (GAPDH) being utilized as the endogenous reference control. The parameters of the amplification for the RT-qPCR analyses were as follows: two min at 50 °C, 20 s at 95 °C, and 40 cycles of 3 s at 95 °C followed by 30 s at 60 °C. All samples were collected in triplicate from 3 culture cells of multi-well culture plates from the seventh day for RT-qPCR. The 2^−ΔΔCt^ formula was applied to calculate the gene expression, and the target genes were normalized against the endogenous GAPDH gene.

### 2.7. Statistical Analysis

All statistical analyses were conducted using IBM SPSS Statistics for Windows, Version 27.0 (SPSS Inc., Chicago, IL, USA). The Kolmogorov–Smirnov normality test and Levene’s homogeneity test were conducted. Two-way ANOVA was undertaken to analyze the expression of Nestin, β-tubulin III, MAP-2, NeuN, and neurites while an independent t-test was used to analyze the ELISA and RT-qPCR results. The threshold for statistical significance was set as * *p* < 0.05; ** *p* < 0.01.

## 3. Results

### 3.1. SCAPs Characterization

Under phase-contrast light microscopy, the SCAPs displayed a fibroblastic-like, spindle-shaped appearance (Figure 1A). Flow cytometry results showed that the SCAPs expressed typical mesenchymal markers for CD73 (100%), CD90 (100%), CD105 (66.1%), and lacked expression of CD45 (1.76%) (Figure 1B). Multiple differentiation assay was conducted to verify the multilineage differentiation potential of SCAPs. SCAPs differentiated into osteoid tissue after 21 days of induction, which was Alizarin Red-positive (Figure 1C). Cultured SCAPs in small-molecule contining neural induction medium expressed morphological changes along with neural markers including β-tubulin III and MAP-2 (Figure 1D). After 27 days of adipogenic induction, adipocytes were identified by Oil Red (Figure 1E). SCAPs formed chondroid tissue after 28 days and were positive for Alcian Blue (Figure 1F).

### 3.2. Cell Viability

The results of the cytotoxic assay among different groups on all time points revealed a high viability of cells seeded on collagen. On the fourth day, the 2D culturing method showed a significantly higher viability of 99.97% ± 0.01% than the 3D spheres (99.33 ± 0.28%) (*p* < 0.01). On the seventh day, no significant difference was observed in the viability among both 2D (99.99% ± 0.02%) and 3D (99.96% ± 0.04%), respectively (*p* > 0.05) (Figure 2).

### 3.3. SCAPs Morphological Changes and Neurites Measurement

SCAP spheres were standardized using the same number of cells in agarose molds. The 3D spheres of similar size were formed after three days of culturing. All spheres ranged from 150 to 200 µm in diameter. The 3D spheres attached well to the collagen I-coated glass, followed by cell migration out of the spheres. The migrating cells’ morphology changed from fibroblast-like cells to neuron-like cells, which were round with long neurites. The typical neuron-like morphology was apparent on day 7.

The neurite number in the 3D spheres was 2.38 ± 0.37 and 2.58 ± 0.44 on days 4 and 7, respectively, and the difference was insignificant (*p* > 0.05). The neurite number in the 2D culture was 1.5 ± 0.14 and 1.5 ± 0.14 on days 4 and 7, respectively, and no significant difference was found (*p* > 0.05). There was significantly more neurite outgrowth in the 3D spheres than that in the 2D culture at both time points (*p* < 0.01) (Figure 3A–C).

The mean length of neurites in the 3D spheres was 64.12 ± 8.02 and 143 ± 8.65, respectively, and the difference was statistically significant (*p* < 0.01). The mean length of neurites in the 2D culture was 51.7 ± 5.7 and 58.8 ± 6.28 on days 4 and 7, respectively, and no significant difference was found (*p* > 0.05). The mean neurite length was significantly different between the 2D and 3D cultures (*p* < 0.05) at both time points (Figure 3A–C).

The total neurite length in the 3D spheres was 1002.38 ± 266.55 and 2880.95 ± 771.7 on days 4 and 7, respectively, and the difference was statistically significant (*p* < 0.01). The total neurite length in the 2D culture was 462.03 ± 22.93 and 530.52 ± 87.75 on days 4 and 7, respectively, and no significant difference was found (*p* > 0.05). No significant difference in the total neurite length at day 4 was found between the 2D and 3D cultures (*p* > 0.05), while the total neurite length was significantly increased in the 3D spheres at day 7 (*p* < 0.01) (Figure 3A–C).

### 3.4. Secretion of BDNF and NGF by SCAP Spheres

After being cultured in α-MEM for seven days, an ELISA assay was carried out to determine the neurotrophic secretion of the 2D and 3D cultures. The level of BDNF expressed by the 2D culture was 118.91 ± 7.13 pg/mL, which was significantly higher than the level secreted in the 3D SCAP spheres (63.68 ± 10.13 pg/mL) (*p* < 0.01). The mean level of NGF-β was 15 ± 0.91 pg/mL in the 2D SCAP culture, which was also significantly higher than that in the 3D SCAP spheres (3.16 ± 1.78 pg/mL) (*p* < 0.01) (Figure 3D).

### 3.5. Immunofluorescence

The expression of Nestin in the 3D spheres was 7.51 ± 1.25 and 9.93 ± 1.5 on days 4 and 7, respectively, and the difference was statistically significant (*p* < 0.01). The expression of Nestin in the 2D culture was 1.1 ± 0.09 and 1.14 ± 0.18 on days 4 and 7, and the difference was not significant (*p* > 0.05). The expression of Nestin at both time points was significantly higher in the 3D spheres than that in the 2D culture (*p* < 0.01) (Figure 4).

The expression of NeuN in the 3D spheres was 1.18 ± 0.18 and 2.8 ± 1.9, respectively, and the difference between both time points was statistically significant (*p* < 0.01). The expression of NeuN in the 2D culture was 0.35 ± 0.024 and 0.21 ± 0.05 on days 4 and 7, respectively, and no significant difference was found between them (*p* > 0.05). At both time points, NeuN expression was significantly increased in the 3D spheres compared to the 2D culture (*p* < 0.01) (Figure 4).

The expression of β-tubulin III in the 3D spheres was 2.41 ± 0.82 and 10.3 ± 1.54, respectively, and the difference between both time points was statistically significant (*p* < 0.01). The expression of β-tubulin III in the 2D culture was 3.12 ± 0.45 and 3.04 ± 0.37 on days 4 and 7, respectively, and no significant difference was found between them (*p* > 0.05). The expression of β-tubulin III was not significantly different between the 2D and 3D cultures at day 4 (*p* > 0.05), while the expression was significantly higher in the 3D spheres compared to the 2D culture at day 7 (*p* < 0.01) (Figure 5).

The expression of MAP-2 in the 3D spheres was 2.83 ± 0.42 and 2.75 ± 0.41, respectively, and the difference between both time points was not significant (*p* > 0.05). The expression of MAP-2 in the 2D culture was 1.7 ± 0.24 and 1.77 ± 0.35 on days 4 and 7, respectively, and no significant difference was found (*p* > 0.05). At both time points, MAP-2 expression was significantly higher in the 3D spheres than that in the 2D culture (*p* < 0.01) (Figure 5).

### 3.6. RT-qPCR

RT-qPCR was carried out to compare the expression of immature and mature neural markers among SCAPs cultured as spheres or monolayer cells on the seventh day. The relative gene expressions of Oct-4, Sox-2, Sox-10, Nestin, β-tubulin III, MAP-2, NeuN, NSE, NFM, BDNF, and NGF were analyzed. SCAP sphere gene expression was compared to the 2D control group, as shown in Figure 6.

Regarding progenitor and immature gene expression, all genes were significantly upregulated (*p* < 0.01) except Sox-10 (*p* > 0.05). The 3D spheres showed 1.7- and 1.5-fold upregulation at the RNA level of Oct-4 and Sox-2, respectively, in comparison to the 2D culture (*p* < 0.01). The expressions of Nestin and β-tubulin III in spheres were 10.5 times and 4.9 times higher than the control monolayer (*p* < 0.01) (Figure 6).

Interestingly, all mature neural markers were significantly upregulated in the 3D culturing environment (*p* < 0.01). With 3D spheres, the relative gene expressions of NeuN and NFM were 6.9- and 2.8-fold higher than the 2D control (*p* < 0.01). The NSE and MAP-2 RNA levels were 27.6- and 29-fold higher in the SCAP sphere group (*p* < 0.01). Furthermore, neurotrophic genes BDNF and NGF were significantly elevated in 3D samples more than 12 times higher than the 2D models (*p* < 0.01) (Figure 6).

## 4. Discussion

Nestin is a known marker of neural stem cells or progenitor cells. It is also constitutively expressed in dental stem cells, such as DPSCs, SCAPs, and periodontal ligament stem cells [8,21]. Since dental stem cells express neural crest-associated markers, it is not difficult to differentiate them into neurons under neural inductive settings both in vitro and in vivo [21,22]. Following neural induction, the early neural markers of neural stem cells/progenitor cells, Nestin and Musashi1, could be downregulated while the markers of immature and mature neurons, β-tubulin III, NeuN, and MAP-2, could be gradually increased in a time-dependent manner [23,24]. Our previous study also demonstrated that neuronal differentiation of dental stem cells exhibits a similar pattern of neural marker expression [17]. However, after long-term culturing and multiple passaging in 2D conditions without any specific induction, MSCs may naturally undergo osteogenic or chondrogenic differentiation with the upregulation of the corresponding genes and proteins while losing their stemness markers and neurogenic potential [25]. In contrast, the 3D sphere conditions may be able to suppress the inherent traits of SCAPs toward osteogenic or chondrogenic differentiation by increasing neurotrophin receptors [26]. In this respect, the formation of 3D spheres could be a viable way for SCAPs to expand before neural induction. In addition, transplantation of 3D spheres of SCAPs may lessen the detrimental effects of the harsh local microenvironment on cell survival and prevent the transplanted cells from differentiating into other lineage fates, causing the formation of non-neural tissue. Interestingly, without any neural induction, 3D sphere-derived DPSCs can spontaneously convert into neuron-like cells positive for neural markers HuC/D and P75 in a serum-free medium [14]. These studies highlighted the importance of 3D culture, neural induction medium, and serum-free medium in the neural differentiation of dental stem cells. In this study, we investigated the solitary effects of the 3D spheres culturing method on SCAPs’ neurogenic potential without using serum-free and/or induction medium.

The subsequent impact of α-MEM with 10% FBS on the SCAP neurogenic potential may affect SCAP neuroprotection and neuroregeneration capacities [25,26]. It was found that the Nestin expression level did not change in 2D culture in α-MEM with 10% FBS from day 4 to 7, while it was significantly increased in the 3D spheres. Our RT-qPCR result agreed with the confocal results and showed a more than ten-fold rise of Nestin in the 3D group on the seventh day compared to the control group. Additionally, the RT-qPCR results are consistent with Völlner, F. and Ernst, W. [27], who found that both immature (Nestin and β-tubulin III) along with the mature (NSE and NFM) neural markers in 3D spheres were significantly upregulated. 3D sphere formation not only maintains the stemness of the embedded MSCs but also leads to the upregulation of several immature and mature neural markers [28,29,30]. Our findings showed a similar result after testing a variety of neural genes. The results showed significantly upregulated Oct-4, Sox-2, MAP-2, NeuN, NGF, and BDNF in SCAP spheres. The results strongly suggest that the formation of 3D spheres can enhance SCAP properties like neural stem cells/progenitor cells in a serum-containing medium. In addition, the 3D culturing model of SCAPs may comprise a mixture of NPCs and neuronal cells, as demonstrated by the expression of early, immature, and mature neuronal markers. However, when SCAPs were cultured in 2D conditions, both immature and mature neural markers were not upregulated in the absence of a neural induction medium, suggesting the critical role of a neural induction medium in monolayer culture.

It was shown that DPSCs and SCAPs cultured in a neurosphere culturing condition (DMEM/F12 (1:1) supplemented with bFGF, 20 ng/mL EGF, and N2 supplement) formulate spheres that possess the characteristics of NSCs [31]. Similarly, several studies have shown that the sphere formation of dental stem cells enhanced their stemness and differentiation ability [28,32,33]. The 3D spheres may provide favorable conditions for neural regeneration via increasing intra-sphere and intracellular communication at the ultrastructural level. Furthermore, cells within the sphere exhibit prominent nucleoli, Golgi apparatus, and an enlarged endoplasmic reticulum, demonstrating elevated metabolic activity [29]. Nevertheless, sphere size is crucial to maintaining a healthy condition. Dissanayaka, W.L. et al. [34] showed that the oxygen diffusion would be compromised beyond 200 μm. Our results express more than 99% vitality in spheres because of the sufficient oxygen diffusion in the generated 150 to 200 µm SCAP spheres.

The migrating cells’ morphology in the 3D spheres was round with long neurites, resembling neuron-like cells. The typical neuron-like morphology was more apparent on day 7. In line with the morphological changes, the neurite number, the mean length of neurites, and the total length of neurites in the 3D spheres were increased from day 4 to 7 and were much higher than those in the 2D culture. These results suggest that 3D sphere-derived SCAPs have superior neurogenic differentiation capacity than 2D cultured cells. The results are corroborated by the previous study, which reported that short-term spheroid formation of ASCs can increase their therapeutic potential for tissue regeneration [13].

Neurotrophins play critical roles in regulating neural cell survival, differentiation, and neurite outgrowth. It was reported that BDNF and NGF-β, the low-molecular-weight proteins of neurotrophins, can significantly increase the neurite length of neural stem cells [35] (pp. 547–550). Neurotrophins initiate signaling cascades by binding with high-affinity tyrosine kinases (Trk) or low-affinity neurotrophin P75 receptors [35]. Multiple studies have shown that dental stem cell-derived spheres express more Trk and P75 neurotrophin receptors than single-cell cultures [26,31,36,37]. Moreover, Wetmore et al. [38] studied the regulation of BDNF within the hippocampus neural tissue and provided evidence for the autocrine effect of BDNF. Ghosh et al. [39] showed that BDNF played an essential role in neural cell survival. Both studies found that the level of BDNF secretion from the neurons increased by applying a stimulus, for example, electrical and/or chemical stimuli [38,39]. In our study, the low BDNF and NGF-β levels shown in the ELISA results could be explained by more efficient utilization, which subsequently promoted neural differentiation of SCAPs and low secretion in the absence of any external stimuli. Our RT-qPCR results confirmed the neural differentiation through the gene upregulation of both BDNF and NGF-β. This suggestion is further supported by the fact that several immature and mature neural markers were highly upregulated, and more typical neuron-like cells were found on day 7 in the 3D conditions.

## 5. Conclusions

This study illustrated that the formation of 3D spheres enhanced the neurogenic potential of SCAPs in terms of neural marker expression and neurite length and number. Signaling cascades initiated by BDNF and NGF-β could be a key contributing factor. The formation of 3D spheres could be a viable way for SCAP expansion and neural differentiation.

## Figures and Tables

**Figure 1 bioengineering-09-00604-f001:**
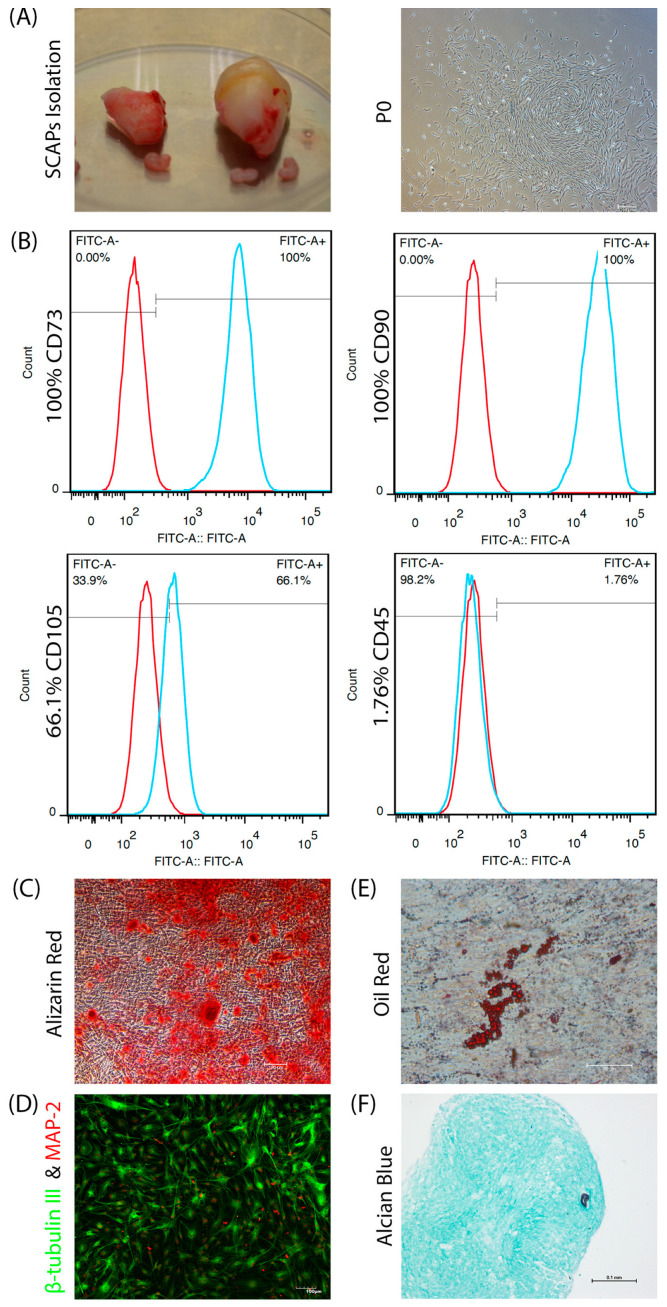
(**A**) Microscopic images for SCAPs isolation; (**B**) Flow cytometric results for SCAPs characterization; Microscopic images for (**C**) osteogenic, (**D**) neurogenic, (**E**) adipogenic, and (**F**) chondrogenic multi-differentiation assay.

**Figure 2 bioengineering-09-00604-f002:**
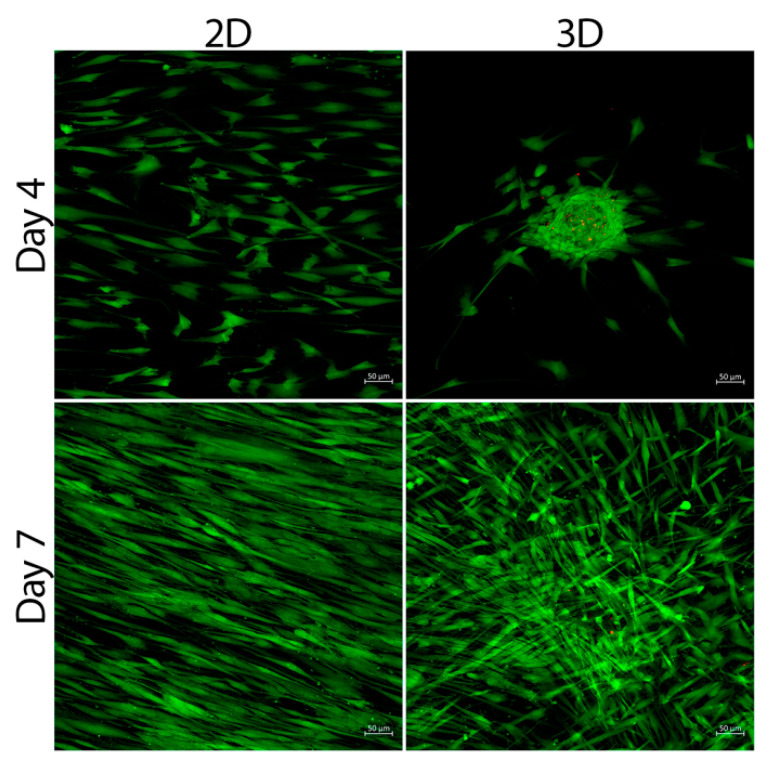
Live and dead staining for 3D SCAP spheres and 2D SCAPs at different time points.

**Figure 3 bioengineering-09-00604-f003:**
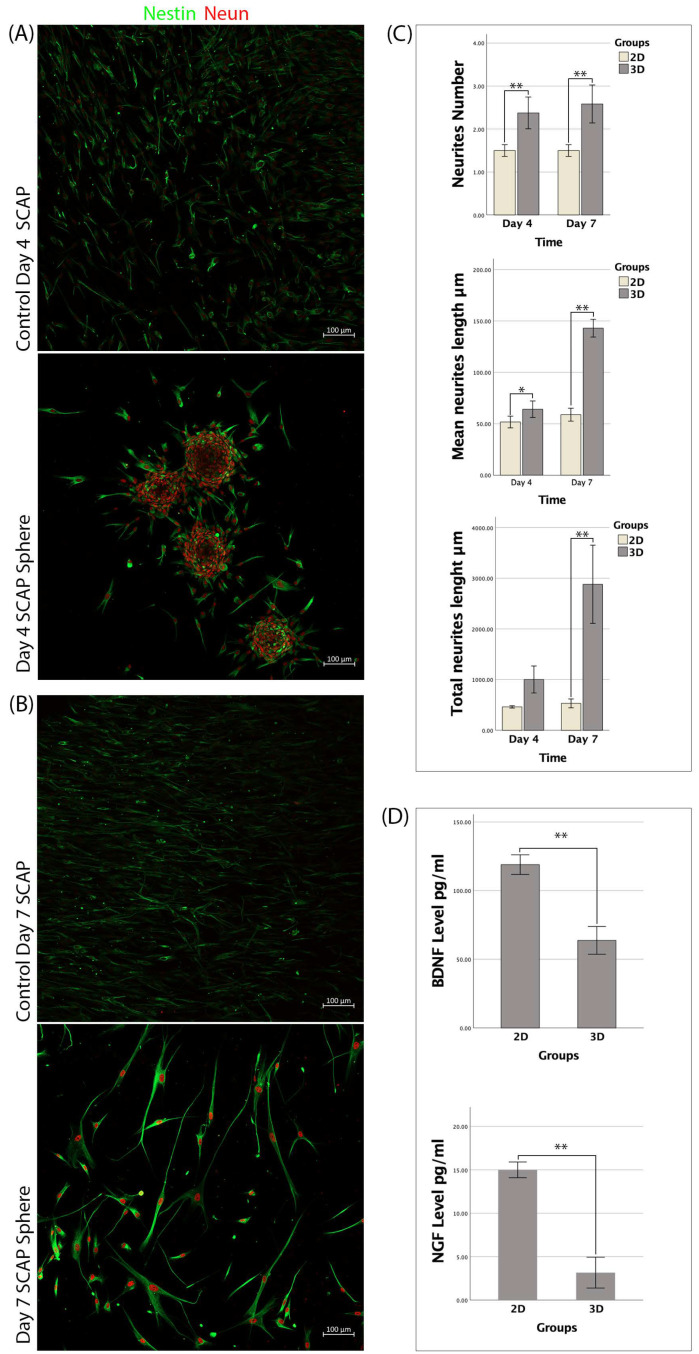
(**A**,**B**) Nestin and NeuN immunofluorescence staining showed changes in cell morphology on days 4 and 7; (**C**) Neurites’ number, mean length, and total length analysis, n = 8; (**D**) ELISA results for the expression of BDNF and NGF in pg/mL. Note (*: *p* < 0.05/**: *p* < 0.001).

**Figure 4 bioengineering-09-00604-f004:**
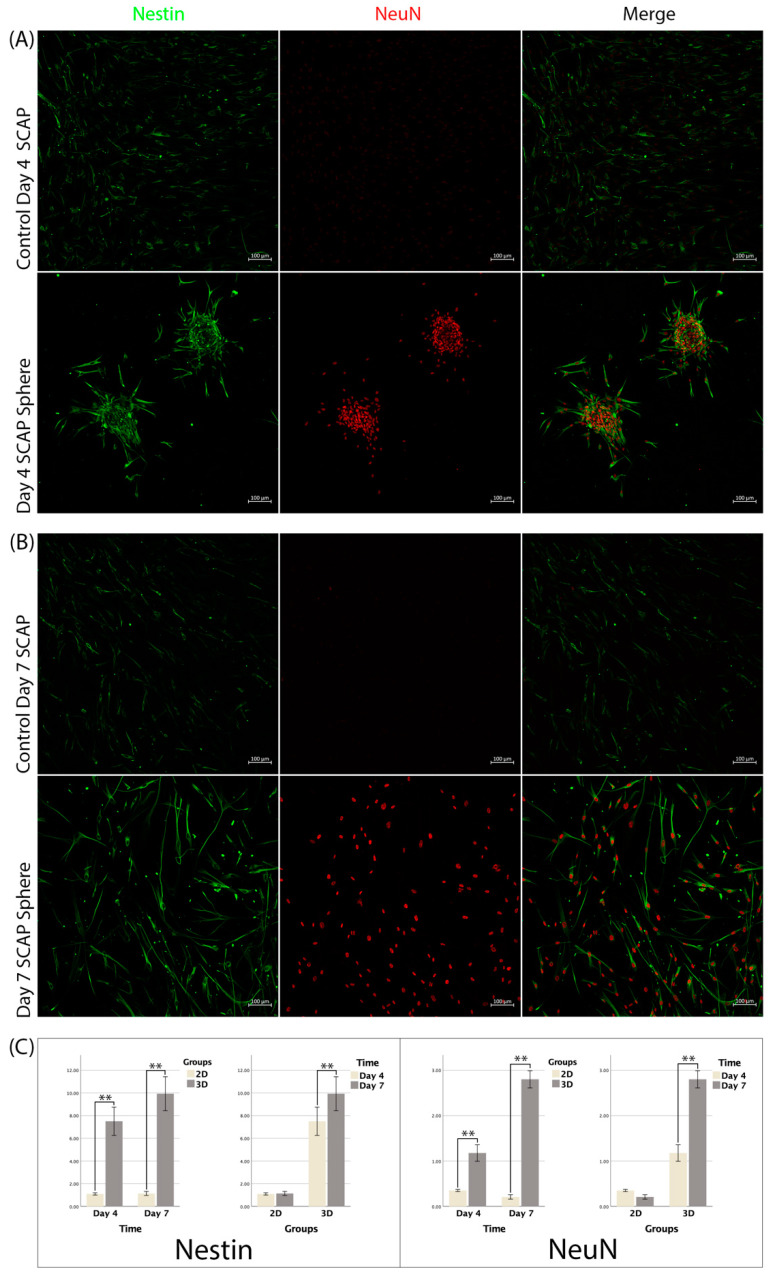
(**A**,**B**) Immunofluorescence staining for 3D SCAP spheres and 2D SCAPs for detecting Nestin and NeuN expression at different time points; (**C**) Neural markers expression analysis of 2D and 3D SCAPs in two time points, n = 12/group (**: *p* < 0.001).

**Figure 5 bioengineering-09-00604-f005:**
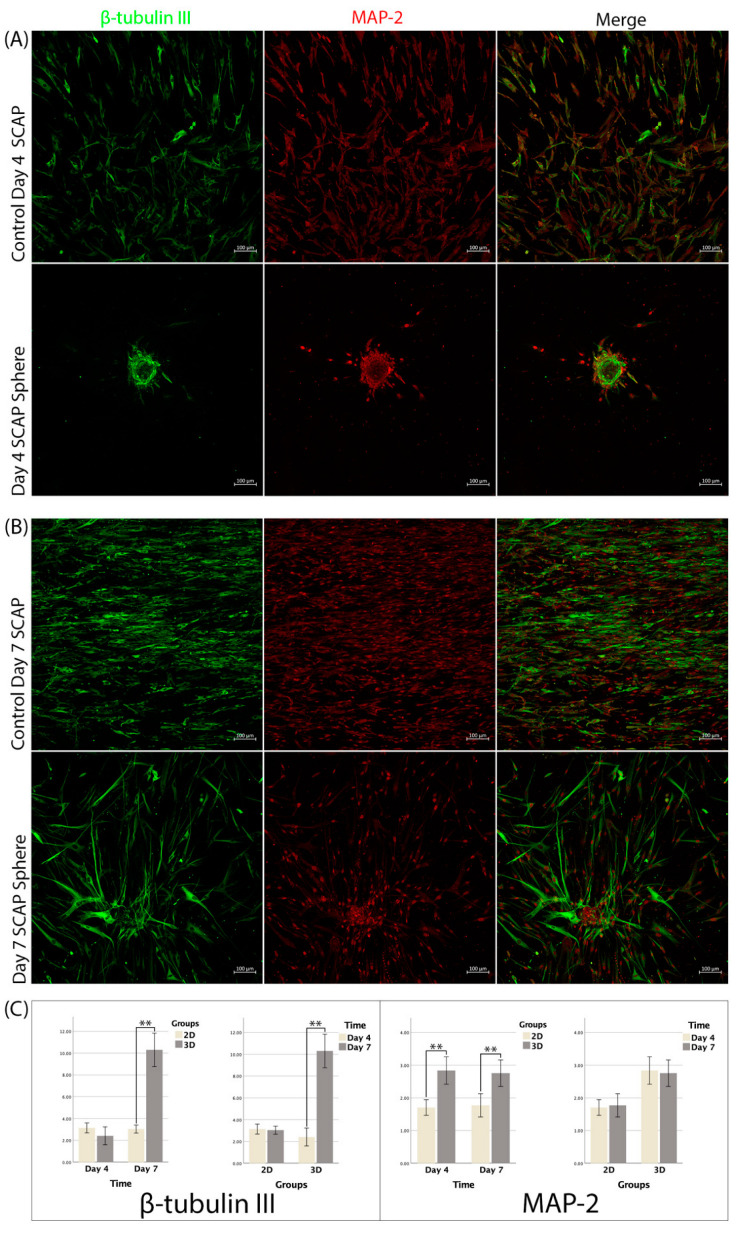
(**A**,**B**) Immunofluorescence staining for 3D SCAP spheres and 2D SCAPs for detecting βIII and MAP-2 expression in different time points; (**C**) Neural markers expression analysis of 2D and 3D SCAPs in two time points, n = 12 (**: *p* < 0.001).

**Figure 6 bioengineering-09-00604-f006:**
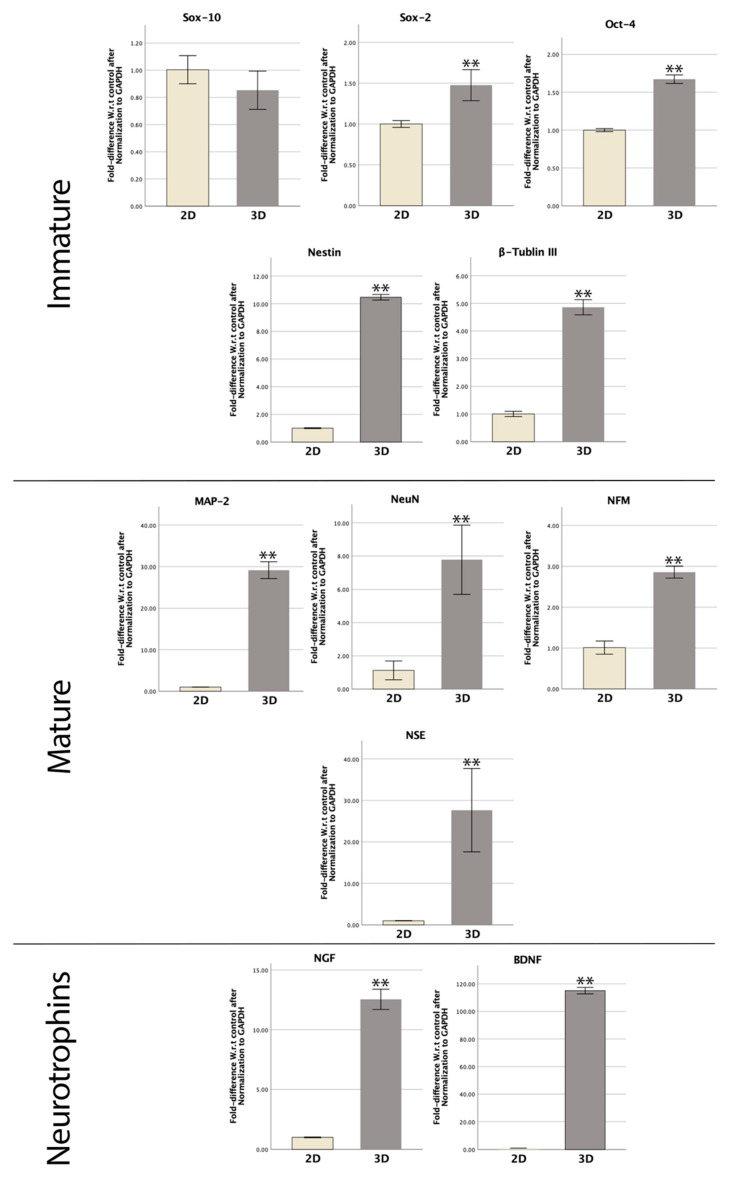
RT-qPCR for control SCAPs and 3D-cultured SCAPs at day 7 for the fold difference with reference to control after normalization to GAPDH of multiple neural markers (Student’s *t*-test, **: *p* < 0.01).

**Table 1 bioengineering-09-00604-t001:** RT-qPCR primer sequences utilized.

Gene		Primer Sequences
Immature	Oct-4	F 5′-GTTGATCCTCGGACCTGGCTA-3′ R 5′-GGTTGCCTCTCACTCGGTTCT-3′
	Sox-2	F 5′-AGTCTCCAAGCGACGAAAAA-3′ R 5′-GCAAGAAGCCTCTCCTTGAA-3′
	Sox-10	F 5′-CCTCACAGATCGCCTACACC-3′ R 5′-CATATAGGAGAAGGCCGAGTAGA-3′
	Nestin	F 5′-CAACAGCGACGGAGGTCTC-3′ R 5′-GCCTCTACGCTCTCTTCTTTGA-3′
	β-tubulin III	F 5′-AGACCTACTGCATCGACAACGAGG-3′ R 5′-GCTCATGGTGGCCGATACCAGG-3′
Mature	MAP-2	F 5′-TTGGTGCCGAGTGAGAAGAA-3′ R 5′-GGTCTGGCAGTGGTTGGTTAA-3′
	NeuN	F 5′-GCGGCTACACGTCTCCAACATC-3′ R 5′-ATCGTCCCATTCAGCTTCTCCC-3′
	NSE	F 5′-GTCCCACGTGTCTTCCACTT-3′ R 5′-TGGGATCTACAGCCACATGA-3′
	NFM	F 5′-GTCAAGATGGCTCTGGATATAGAAATC-3′ R 5′-TACAGTGGCCCAGTGATGCTT-3′
Neurotrophins	BDNF	F 5′-TAACGGCGGCAGACAAAAAGA-3′ R 5′- TGCACTTGGTCTCGTAGAAGTAT-3′
	NGF	F 5′- TGTGGGTTGGGGATAAGACCA-3′ R 5′- GCTGTCAACGGGATTTGGGT-3′
Control	GAPDH	F 5′-TGTCTCCTCCGACTTCAACA-3′ R 5′-GCCATGTGGGCCATGAGGT-3′

## Data Availability

All the datasets are available online and have been duly referenced within the text.

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
