# Peer review of "Formation of Three-Dimensional Spheres Enhances the Neurogenic Potential of Stem Cells from Apical Papilla"

_bioengineering, 2022, doi:10.3390/bioengineering9110604_

Round 1

Reviewer 1 Report

The use of 3D spheroids is becoming more widespread, especially in the creation of neurons and neuronal networks in vitro. The manuscript presents an interesting approach that is relevant. At the same time, the tests carried out are limited to only one method, which is not enough to present and verify the results, and the novelty content is not enough either. In any case, many new measurements would be necessary in order for the manuscript to meet the standard of the journal. Such would be the viability and metabolism measurements, More precise sections with immune and histochemistry. Comparison of gene expression patterns could actually show the effectiveness of the method, etc. In its current form, the manuscript is not suitable for publication in the journal.

Author Response

Point-by-point responses to the reviewers’ comments
First, we thank the reviewers for the insightful and constructive comments. We have revised our manuscript accordingly.
Responses to Reviewer #1:
The use of 3D spheroids is becoming more widespread, especially in the creation of neurons and neuronal networks in vitro. The manuscript presents an interesting approach that is relevant. At the same time, the tests carried out are limited to only one method, which is not enough to present and verify the results, and the novelty content is not enough either. In any case, many new measurements would be necessary in order for the manuscript to meet the standard of the journal. Such would be the viability and metabolism measurements, More precise sections with immune and histochemistry.
Comparison of gene expression patterns could actually show the effectiveness of the method, etc. In its current form, the manuscript is not suitable for publication in the journal.

Response: Thank you very much for the comment.
In our previous study, we fabricated 3D spheroids of dental pulp cells (DPCs), which were 200 μm in diameter. LIVE/DEAD cytotoxicity assay revealed a high cell viability and turnover in DPC spheroids at 7 days, and no significant cell death was observed within the center of the spheroids [1]. In this study, the 3D SCAPs spheres were first generated, which were 150 to 200 μm in diameter. Since SCAPs have higher proliferative capacity than DPCs [2], it is reasonable to foresee a high cell viability of SCAPs in the 3D SCAPs spheres. In addition, the cells continuous migration out of the spheres also indicated the cells' viability.
This is why cell viability was not assessed. We conducted immunocytochemistry to assess the neurogenic markers since our experiments were performed in vitro. The immune and histochemistry study as suggested by the reviewer will be conducted in the future in vivo study. We agree that comparing the gene expression is essential. In fact, we had tried several times to extract RNA for the spheres, however, even with 2000 spheres, the concentration of RNA was too low to perform qRT-PCR. It is why the immunofluorescence signals were measured to reveal the protein expression among different groups. We are still optimizing the method for RND extraction.
References:
1. Dissanayaka, WL.; Zhu, L.; Hargreaves, KM.; Jin, L.; Zhang, C. In vitro analysis of scaffold-free prevascularized microtissue spheroids containing human dental pulp cells and endothelial cells. J Endod 2015 41(5), 663-70.
2. Sonoyama, W.; Liu, Y.; Fang, D.; Yamaza, T.; Seo, BM.; Zhang, C.; Liu, H. Gronthos S, Wang CY, Wang S, Shi S. Mesenchymal stem cell-mediated functional tooth regeneration in swine. PLoS One 2006, 1(1), e79,
https://doi.org/10.1371/journal.pone.0000079.

Reviewer 2 Report

In this study, the authors compared the SCAPs from human dental tissue in 2D and 3D cultures and concluded that the formation of 3D spheres enhanced the neurogenic potential of SCAPs for the treatment of neural diseases. The study determined changes in cell morphology, neural marker expression (Nestin, β-III-tubulin, NeuN, and MAP-2), neurite outgrowth, and secretion of BDNF and NGF-β. Despite the reported positive effects on the above-mentioned aspects, there is a major concern regarding the functionality of the SCAP-sphere derived neural cells.

Major concerns:

1.             Since the cells in 3D express mature neuronal markers of NeuN and Map2,the functions of SCAP-sphere derived neurons should be further characterized by electrophysiological study to show that they can elicit action potentials and form synapses.

2.             It has been reported that BDNF and NGF-β significantly increased the neurite length. Thus, it is not clear why cells with a decreased secretion of BDNF and NGF-b in 3D culture compared to 2D culture have a longer neurite length.

3.             There is no direct experimental data from this study to support Figure 5. Thus, it is not appropriate to include in this manuscript.

Minor comments:

1.             Please define SCAP in the abstract (Page 1 line 15)

2.             there should be spaces to separate each images in Figure 2A-B.

3.             there is no mention of the sample size for Figures 2-4. Are all 7 samples were included in these figures?

Author Response

Point-by-point responses to the reviewers’ comments
First, we thank the reviewers for the insightful and constructive comments. We have revised our manuscript accordingly.
Responses to Reviewer #2:
Major concerns:
Q1. Since the cells in 3D express mature neuronal markers of NeuN and Map2, the functions of SCAP-sphere derived neurons should be further characterized by an electrophysiological study to show that they can elicit action potentials and form synapses.

Response: Thank you very much for your advice.
Since the aim of this study was to investigate whether the formation of 3D spheres can promote SCAPs neurogenic potential, we only cultured the spheres for 7 days. We found that the immature and mature neural markers were upregulated in the 3D spheres of SCAPs, and the induced cells may comprise a mixture of NPCs and neuronal cells, as demonstrated by the expression of early, immature, and mature neuronal markers. Because the action potentials
and synapse formation are the functions of mature neurons, the  electrophysiological study may not be appropriate for the mixture of NPCs and neuronal cells. That is why it was not performed. We will definitely consider conducting further characterization by electrophysiological study in the future.

Q2. It has been reported that BDNF and NGF-β significantly increased the neurite length. Thus, it is not clear why cells with a decreased secretion of BDNF and NGF-b in 3D culture compared to 2D culture have a longer neurite length.

Response: Thank you very much for the comment. Neurotrophins, such as BDNF and NGF-β, initiate signaling cascades by binding with highaffinity
tyrosine kinases (Trk) or low-affinity neurotrophin P75 receptors [1]. Multiple studies showed that dental stem cell-derived spheres express more Trk and P75 neurotrophin receptors than single-cell culture [2,3,4,5]. Therefore, BDNF and NGF-β secreted by SCAPs under 3D conditions could be more efficiently utilized and subsequently promote neural differentiation of SCAPs, leading to lower BDNF and NGF levels in the culture media. Therefore, the longer neurites in the 3D spheres in this study than that in the 2D culture could possibly be a result of local consumption of both neurotrophins.
References
1. Purves, D.; Augustine, GJ.; Fitzpatrick, D.; Hall, W.; LaMantia, A-S.; White L. Neuroscience, Sixth ed.; Oxford University Press: New York, NY, USA, 2018; pp. 547-550, ISBN 9781605353807, 1605353809.
2. Luzuriaga, J.; Pineda, JR.; Irastorza, I.; Uribe-Etxebarria, V.; García-Gallastegui, P.; Encinas, JM.; Chamero, P.; Unda, F.; Ibarretxe, G. BDNF and NT3 Reprogram Human Ectomesenchymal Dental Pulp Stem Cells to Neurogenic and Gliogenic Neural Crest Progenitors Cultured in Serum-Free Medium. Cell Physiol Biochem
2019, 52(6), 1361-1380, https://doi.org/10.33594/000000096.
3. Abe, S.; Hamada, K.; Miura, M.; Yamaguchi, S. Neural crest stem cell property of apical pulp cells derived from human developing tooth. Cell Biol Int 2012, 36(10), 927-36, https://doi.org/10.1042/CBI20110506.
4. Solis-Castro, OO.; Boissonade, FM.; Rivolta, MN. Establishment and neural differentiation of neural crestderived stem cells from human dental pulp in serum-free conditions. Stem Cells Transl Med 2020, 9(11), 1462-
1476, https://doi.org/10.1002/sctm.20-0037.
5. Widera, D.; Zander, C.; Heidbreder, M.; Kasperek, Y.; Noll, T.; Seitz, O.; Saldamli, B.; Sudhoff, H.; Sader, R.; Kaltschmidt, C.; Kaltschmidt, B. Adult palatum as a novel source of neural crest-related stem cells. Stem Cells 2009, 27(8), 1899-910, https://doi.org/10.1002/stem.104.

Q3. There is no direct experimental data from this study to support Figure 5. Thus, it is not appropriate to include in this manuscript.

Response: Thank you very much for your comments.
We removed the figure from the revised manuscript.

Minor concerns:
Q1. Please define SCAP in the abstract (Page 1 line 15)
Response: Thank you very much for your comments.
We added SCAP definition in the revised manuscript as instructed.

Q2. there should be spaces to separate each images in Figure 2A-B.
Response: Thank you for your advice.
Spaces have been placed in the revised manuscript as instructed.

Q3. there is no mention of the sample size for Figures 2-4. Are all 7 samples were included in these figures?
Response: Thank you very much for your comment.
We isolated SCAPs from 7 patients and characterized all of them. We only used SCAPs which were isolated from one patient in the following experiment. The sentence (Page 2, line 91) was revised as follows: SCAPs were isolated from the freshly extracted third molars of a 20-yearold female patient. The sample size was added to the figure legends.

Round 2

Reviewer 2 Report

I would recommend that the authors conduct the electrophysiological studies at at a more mature time point (e.g., 12-14 days in vitro differentiation) and the gene expression analysis as suggested by Reviewer #1.  If it is difficult to collect sufficient RNA for bulk RNA-seq analysis, single-cell RNA-seq is an option.

Author Response

Responses to Reviewer #Round 2:

I would recommend that the authors conduct the electrophysiological studies at a more mature time point (e.g., 12-14 days in vitro differentiation) and the gene expression analysis as suggested by Reviewer #1.  If it is difficult to collect sufficient RNA for bulk RNA-seq analysis, single-cell RNA-seq is an option.

Response:

Thank you very much for the recommendation.
We used another method (Trizol RNA extraction) for our samples cultured in Rat tail collagen I to isolate the mRNA and conducted the RT-qPCR for several genes. Furthermore, we performed live and dead analyses and added the results to the manuscript. On the other hand, we will do an electrophysiological study in our subsequent article because we are organizing a collaboration with another university, and that will take time.

Round 3

Reviewer 2 Report

The revised manuscript addressed this reviewer's comments.